# Plasma Lysophosphatidylcholine Levels Correlate with Prognosis and Immunotherapy Response in Squamous Cell Carcinoma

**DOI:** 10.3390/ijms26157528

**Published:** 2025-08-04

**Authors:** Tomoyuki Iwasaki, Hidekazu Shirota, Eiji Hishinuma, Shinpei Kawaoka, Naomi Matsukawa, Yuki Kasahara, Kota Ouchi, Hiroo Imai, Ken Saijo, Keigo Komine, Masanobu Takahashi, Chikashi Ishioka, Seizo Koshiba, Hisato Kawakami

**Affiliations:** 1Department of Medical Oncology, Tohoku University, Sendai 980-8575, Japan; tomoyuki.iwasaki.s8@dc.tohoku.ac.jp (T.I.); yuki.kasahara.d8@tohoku.ac.jp (Y.K.); kota.ouchi.b3@tohoku.ac.jp (K.O.); hiroo.imai.d8@tohoku.ac.jp (H.I.); ken.saijo.d6@tohoku.ac.jp (K.S.); keigo.komine.e7@tohoku.ac.jp (K.K.); masanobu.takahashi.a7@tohoku.ac.jp (M.T.); chikashi@tohoku.ac.jp (C.I.); kawakami_h@tohoku.ac.jp (H.K.); 2Advanced Research Center for Innovations in Next-Generation Medicine, Tohoku University, Sendai 980-8573, Japan; eiji.hishinuma.e7@tohoku.ac.jp (E.H.); matsukawa@megabank.tohoku.ac.jp (N.M.); seizo.koshiba.b3@tohoku.ac.jp (S.K.); 3Tohoku Medical Megabank Organization, Tohoku University, Sendai 980-8573, Japan; 4Department of Integrative Bioanalytics, Institute of Development, Aging and Cancer, Tohoku University, Sendai 980-8575, Japan; shinpei.kawaoka.c1@tohoku.ac.jp

**Keywords:** squamous cell carcinoma, metabolome, proteome, cancer inflammation, immune checkpoint inhibitor

## Abstract

Cancer is a systemic disease rather than a localized pathology and is characterized by widespread effects, including whole-body exhaustion and chronic inflammation. A thorough understanding of cancer pathophysiology requires a systemic approach that accounts for the complex interactions between cancer cells and host tissues. To explore these dynamics, we employed a comprehensive metabolomic analysis of plasma samples from patients with either esophageal or head and neck squamous cell carcinoma (SCC). Plasma samples from 149 patients were metabolically profiled and correlated with clinical data. Among the metabolites identified, lysophosphatidylcholine (LPC) emerged as the sole biomarker strongly correlated with prognosis. A significant reduction in plasma LPC levels was linked to poorer overall survival. Plasma LPC levels demonstrated minimal correlation with patient-specific factors, such as tumor size and general condition, but showed significant association with the response to immune checkpoint inhibitor therapy. Proteomic and cytokine analyses revealed that low plasma LPC levels reflected systemic chronic inflammation, characterized by high levels of inflammatory proteins, the cytokines interleukin-6 and tumor necrosis factor-α, and coagulation-related proteins. These findings indicate that plasma LPC levels may be used as reliable biomarkers for predicting prognosis and evaluating the efficacy of immunotherapy in patients with SCC.

## 1. Introduction

Owing to their unique biological characteristics, cancer cells develop altered metabolic pathways, which allow them to support uncontrolled cell proliferation [1,2]. Consequently, the host organism responds to these abnormal cells in various ways, such as through the immune response, which significantly alters the metabolism of normal tissues [3,4,5]. In this sense, cancer is not a localized disease that affects only the tumor site but a systemic disease that causes metabolic abnormalities. Owing to significant abnormalities in metabolic pathways, a condition similar to systemic wasting syndrome develops in the majority of patients with advanced cancer [6,7,8]. Targeting these metabolic abnormalities is a promising approach to personalized medicine, potentially leading to novel therapeutic targets.

In recent years, personalized medicine based on the next-generation sequencing molecular diagnosis of cancer has been established and is now widely implemented in routine clinical practice [9,10]. In cancers such as adenocarcinoma, identifying specific genetic mutations and developing molecularly targeted therapies have significantly improved treatment outcomes [11]. However, in squamous cell carcinoma (SCC), the utility of molecular diagnostics based on genomic alterations is limited, creating a gap in the development of personalized treatment. As the prognosis for advanced head and neck and esophageal cancers is poor, alternative diagnostic approaches and novel therapeutic targets are urgently needed [12]. At diagnosis, many patients with SCC present with cachexia, which is a serious deterioration of their general condition that negatively influences the prognosis and complicates treatment [12,13]. Cancer-related metabolic disruptions extend beyond glycolysis, involving abnormalities in amino acid and lipid metabolism that impair systemic physiological functions [14,15,16]. To address these challenges, metabolomic and proteomic analyses have emerged as promising alternatives to genomic approaches [17,18]. In particular, recent technological advances have enabled the detection of a wider range of metabolites with higher sensitivity [19]. These altered metabolites are implicated not only in the direct regulation of tumor proliferation but also in changes in the tumor microenvironment and activation of immune cells. Elucidating the metabolic cycles and biochemical- and protein-level changes associated with SCC may contribute to the discovery of potential biomarkers, providing insight into the underlying disease mechanisms to aid diagnosis and therapeutic targets. These complementary strategies may help in developing more effective and personalized treatment options.

In this study, clinical information and multi-omics data, including the comprehensive metabolome and proteome data of patients with SCC, were analyzed to identify novel biomarkers. Lysophosphatidylcholine (LPC), a metabolite identified from clinical markers correlating with prognosis, was associated with chronic inflammation and with the efficacy of immunotherapy in patients with cancer. Alterations in metabolic pathways in patients with cancer are not only biomarkers for predicting prognosis but also have the potential to elucidate systemic metabolic pathways and clinical conditions, as well as target molecules in drug discovery.

## 2. Results

### 2.1. Correlation Between Clinical Data and Prognosis in Patients with SCC

To perform a comprehensive plasma metabolomic analysis of patients with SCC, the plasma samples of 149 patients undergoing standard chemotherapy at Tohoku University Hospital were analyzed. Table 1 presents the baseline characteristics of the patient cohort. Approximately half of the patients were diagnosed with esophageal SCC, whereas the remainder had head and neck SCC. The majority of patients (85%) were older men. All patients were undergoing or had undergone chemotherapy for stage IV disease, with measurable tumor lesions identified through computed tomography according to the response evaluation criteria in solid tumor guidelines [20]. None of the patients exhibited concurrent infections. Plasma samples were collected at various treatment lines during chemotherapy: 21 samples were obtained before chemotherapy initiation, 64 during first-line treatment, 46 during second-line treatment, and 18 during third-line or later treatments. All samples were collected during the withdrawal period of chemotherapy. Clinical data, such as patient conditions or characteristics, including body mass index, history of body weight loss within the 6 months before chemotherapy, and laboratory data, were recorded before the initiation of first-line treatment. In addition, scores related to nutritional status, such as the Glasgow prognostic score (GPS), controlling nutritional status (CONUT) score, and neutrophil-to-lymphocyte ratio, were calculated [21,22,23]. For all patients, survival data were monitored throughout treatment. The median OS was 22.5 months for esophageal SCC and 21.7 months for head and neck SCC, with nearly identical Kaplan–Meier survival curves for both groups (Figure 1A). Univariable Cox proportional hazards analysis was employed to evaluate correlations between patient prognosis and various clinical factors (Figure 1B). Among the factors examined, only a GPS of ≥1 showed a significant correlation with prognosis (hazard ratio (HR), 1.52; 95% confidence interval (CI), 1.02–2.25). Other variables, including age, sex, severe weight loss, performance status at the first visit, tumor size, number of metastatic sites, and laboratory test results, did not have a significant prognostic effect. These findings identified the GPS as a significant prognostic factor in this cohort, prompting further stratified analyses focusing on this metric.

### 2.2. Identification of Prognostic Factors Based on Targeted Metabolome Data via Weighted Gene Correlation Network Analysis (WGCNA)

Targeted metabolomic analysis of the plasma samples was conducted, quantifying the levels of 635 metabolites. Metabolite data were subjected to WGCNA to identify correlations between metabolite levels and clinical parameters. WGCNA identified 18 distinct modules, each representing clusters of metabolites with correlated levels. The clustering dendrogram and details of these modules are shown in Figure 2A. The Pearson correlation coefficients for each module and clinical parameter were calculated and visualized in a heatmap to evaluate the relationships between these modules and the clinical data (Figure 2B). The module eigengene, representing the principal component summarizing the metabolite profile of each module, was used to assess correlations with clinical parameters. The analysis included samples collected across all treatment lines. No significant correlations were observed between samples collected in the first- and second-line treatments or later. Among the modules, module 6 exhibited the strongest negative correlation with the GPS, which is strongly correlated with patient prognosis (r = −0.42, *p* = 3 × 10^−8^). Appendix A presents correlations between other clinical data and individual modules. Notably, module 6 did not show correlations with other clinical information such as age, performance status (PS), G8 score, tumor size, or number of metastatic organs. These findings indicate that module 6, which most strongly correlated with the GPS, warrants further investigation in the context of prognosis.

### 2.3. LPC in Module 6 Shows a Correlation with Prognosis

Next, the individual metabolites within module 6 that exhibited the strongest correlation with the GPS were investigated. Module 6 contained 10 LPCs, 8 phosphatidylcholine (PC), and 3 cholesteryl ester (CE) metabolites, while our comprehensive metabolome analysis profiled a total of 11 LPCs, 68 PCs, and 11 CEs. Most of the measurable LPCs (10 out of 11) were included in module 6. The correlation between these metabolites and the GPS is presented in Table 2, along with comparisons with data from a healthy cohort (HC). The HC data were obtained from the ToMMo data analyzed by the same platform and utilized as a control group. The metabolite levels in the GPS 0 group for LPCs, PCs, and CEs were comparable with the average values observed in the HC. All 10 LPC metabolites had significantly lower levels in the GPS 1 and 2 groups than in the GPS 0 group, and a clear negative correlation was observed. Representative LPC fractions (16:0, 18:0, 18:0, and 18:1) are shown in Figure 3A. In addition, significant correlations for PC and CE metabolites were observed, though the differences were not as remarkable as for LPCs (Table 2). Within the metabolites of module 6, the LPC fraction displayed the strongest and most consistent correlation with the GPS.

Then, the correlation between individual metabolites within module 6 and patient prognosis was evaluated by univariate Cox proportional hazards analysis (Figure 3B). Although samples were collected from patients at various treatment stages (Figure 3C), no significant differences in LPC levels were observed across treatment lines. This suggests that LPC levels may not serve as a marker of disease progression but may instead reflect host–tumor interactions rather than treatment stage. Because patients with a GPS of 0 and a good prognosis had LPC levels similar to those of the HCs, the relationship between the total LPC value and OS was then investigated, focusing on the low-LPC group.

### 2.4. Kaplan–Meier Survival Analysis Based on LPC Levels

Kaplan–Meier survival curves were constructed to evaluate the prognostic significance of LPC levels. Patients were divided into two groups based on the median LPC level. The analysis revealed significantly shorter OS in the low-LPC group (18.6 months) than in the high-LPC group (28.4 months; HR 1.56, 95% CI 1.05–2.30, *p* = 0.03; Figure 4A). When stratified by cancer type, the trend was more pronounced in patients with head and neck cancer than in those with esophageal cancer (Figure 4B,C). No significant difference was indicated in the OS for esophageal cancer; however, when the cutoff value was lowered, a significant difference similar to that for head and neck cancer was observed (Appendix A). Furthermore, when weighted analysis was performed on significant LPC fractions, the difference in OS was even greater. All patients in this cohort received chemotherapy, and it is likely that the therapeutic response significantly influenced OS. Recent advancements in immune checkpoint inhibitor (ICI) therapies have extended the OS of patients with cancer [24]. The study cohort included patients treated both before and after ICI approval. Therefore, the patients were stratified by ICI treatment history. No significant association was observed between LPC levels and OS among patients who did not receive ICI treatment (Figure 4D). These results suggest a relationship between LPC levels and ICI efficacy (Figure 4E). Recent clinical trials of ICIs have shown that they are more effective in patients who receive them as a first-line treatment [25]. For this reason, only those patients who received ICIs as first-line treatment were compared. A significant correlation was observed between LPC values and OS (Figure 4F); however, no difference in progression-free survival was detected. This result can be explained by the clinical results of ICIs [26]. Patients with low LPC levels exhibited a diminished survival benefit from ICI treatment, suggesting that reduced LPC levels may be a biomarker for poor responsiveness to ICI therapy.

### 2.5. Association of Decreased LPC with Biological Processes and Inflammatory Pathways

Our results indicate that low LPC levels are associated with poor patient prognosis and predictive of responsiveness to ICI therapy. Moreover, the biological processes in patients with low LPC levels were comprehensively evaluated by proteome analysis. A total of 32 randomly selected samples were used for the proteome analysis, quantifying the expression profiles of 936 proteins, and the samples were divided into two groups based on the median LPC level. A total of 380 proteins that could be quantified in more than 80% of cases were used for statistical analysis. Fifty-four proteins exhibited significantly different expression levels between patients with high and low plasma LPC levels. In the low-LPC group, 11 proteins were upregulated, whereas 43 were downregulated (two-tailed *p*-values < 0.1). The 30 proteins that differed most significantly between the two groups are shown in a heatmap (Figure 5A). Several proteins associated with inflammation, such as C-reactive protein (CRP), serum amyloid A protein, and leucine-rich alpha-2 glycoprotein, were detected at high levels in the low-LPC group.

Pathway analysis, an approach focusing on groups of proteins that share common biological functions, pathways, or diseases, was conducted to compare the high- and low-LPC groups. Figure 5B shows the 25 pathways listed in order of decreasing q value. The Reactome database was used to analyze proteins indicating significant changes, revealing that pathways related to inflammation, innate immunity, and the coagulation system were the most common, signifying that a decrease in LPCs is linked to immune system activation. This result was consistent with the original analytical pattern when a stricter significance threshold of *p* < 0.05 was applied.

Then, the relationship between LPC levels and inflammation was investigated. The cytometric bead array technique was employed to comprehensively quantify the cytokines and chemokines in plasma. Figure 5C shows that the low-LPC group exhibited significantly higher levels of IL-6, IL-10, and TNF-α. In contrast, CCL2 and CXCL8 levels were significantly increased in the high-LPC group. The differences in plasma LPC levels and their associated cytokine profiles suggest distinct inflammatory responses in the high- and low-LPC groups. Additionally, no significant differences were observed in IL-2, IL-4, IL-17A, IFN-γ, CCL5, CXCL9, and CXCL10 levels. These findings provide further evidence that low LPC levels may contribute to the activation of inflammatory pathways, potentially influencing patient prognosis.

## 3. Discussion

Cancer, a systemic disease, affects the entire host organism, causes systemic metabolic dysregulation, suppresses the immune system, promotes chronic inflammation, and instigates a wide range of abnormalities in the nervous and endocrine systems [5,6,7]. Plasma metabolome analysis offers a unique perspective on systemic responses beyond tumor-specific alterations, including those of normal organs and the immune system [3,4,8]. In this study, the association between clinical factors and metabolomic profiles was investigated in patients with either esophageal or head and neck SCC, stratifying patients based on prognostic differences. Comprehensive metabolomic profiling of patient plasma identified over 600 metabolites, with LPC emerging as the sole metabolite significantly associated with prognosis. All patients in this cohort received chemotherapy, and analysis of clinical data revealed that systemic chronic inflammation was the main factor that affected prognosis rather than tumor size or previous weight loss, a sign of cachexia. Low LPC levels were associated with shorter OS following immunotherapy, particularly with first-line ICIs. Proteomic analysis revealed associations between plasma LPC levels and inflammatory markers, in addition to coagulation factors, suggesting the activation of the innate immune system. The analysis of cytokines and chemokines correlated LPC levels with two distinctive inflammatory patterns; plasma LPC was negatively correlated with IL-6, IL-10, and TNF-α, but positively correlated with CCL2 and CXCL8. In this study, cutoff values were set at the median (LPC = 155) to simplify analysis. However, the greatest difference in prognosis was observed in the lower quintile of LPC values (LPC = 105; Appendix A). At this cutoff, significant differences in OS were also observed in both the head and neck and the esophageal SCC groups. The plasma LPC level holds promise as a biomarker for predicting prognosis and treatment response to immunotherapy. It may also indicate chronic inflammation and coagulation abnormalities, aiding in overall patient management. Elucidating the mechanisms underlying LPC reduction in patients with poor prognosis could lead to identifying novel therapeutic targets.

LPC, a key component of cell membranes, also serves as an energy source and a bioactive lipid [27]. Ongoing research explores the complex metabolic pathways involved in LPC degradation and synthesis [28]. Plasma LPC is derived primarily from the degradation of PC present in plasma low-density lipoprotein (LDL) and high-density lipoprotein (HDL) cholesterol. Subsequently, LPC is metabolized into lysophosphatidic acid (LPA), a potent lipid mediator implicated in various physiological processes, including angiogenesis and fibrosis. In the current analysis, a decrease in plasma LPC level was not accompanied by a reduction in either its source (LDL/HDL) or its degradation product (LPA) (Appendix A). Although a significant difference was detected in PC levels, the magnitude of this difference (approximately 10%) was considered relatively modest. Therefore, our results showed that only the LPC levels decreased in the lysophospholipid metabolic pathway.

Many studies have demonstrated a link between plasma LPC and cancer development or progression. Plasma LPC levels are reduced in patients with colorectal, lung, bile duct, and cervical cancers compared with healthy individuals, proposing the potential of LPC as a marker for the early diagnosis of cancer [29,30,31,32]. Other reports have suggested that LPC may serve as a marker for predicting recurrence after cancer surgery and that it may be related to the risk of future cancer development in healthy individuals [33,34]. These reports are consistent with those seen in half of the patients in this study with a poor prognosis, but the detailed mechanism remains unknown. The relationship between the cancer-specific metabolic cycle and LPC levels has been reported in tumor tissues and in in vitro studies. Schmidt et al. reported a marked depletion of LPC in tumor tissue sections of head and neck SCC [35]. Kamphorst et al. reported that only LPC is required for cell membrane replication during cancer cell proliferation in a hypoxic state [36]. In vivo lipid scavenging may also occur throughout the body. If LPC depletion in vivo only suppresses the cancer metabolic cycle, it may be a therapeutic target.

A comparative analysis of the associations between LPC levels, the proteome, and cytokines revealed that most related substances participated in inflammatory processes. Patients with SCC often exhibit high CRP levels, enhanced inflammatory responses, and clinical fever. Chronic inflammation has been widely reported as a hallmark of SCC, with persistent stimuli such as smoking, alcohol consumption, and viral infections contributing to its development [37]. Our findings indicate that patients with more severe cancer-related chronic inflammation have significantly lower LPC levels, correlating with poorer prognosis. LPC is a bioactive lipid with well-documented involvement in inflammation [38,39]. In this study, reduced LPC levels were associated with systemic inflammation in cancer patients. Notably, despite this heightened inflammatory state, ICI therapy was less effective, suggesting a complex interplay between inflammation and antitumor immunity. LPC is classically known for its pro-inflammatory functions, serving as a mediator that activates immune cells and vascular endothelial cells. It is generated through phospholipase A2 activation in response to cellular damage or oxidative stress and is elevated in various inflammatory conditions such as tissue injury, atherosclerosis, infections, and autoimmune diseases. Conversely, LPC also exhibits anti-inflammatory effects under certain conditions. It has been reported to induce immunosuppressive cell populations, such as regulatory T-cells and myeloid-derived suppressor cells, and to promote anti-inflammatory cytokines like IL-10 [38,40]. In chronic inflammatory diseases—including infections, drug-induced pneumonitis, and lipopolysaccharide (LPS)-induced shock—reduced LPC levels have been observed [39,41,42,43]. In cancer, such reductions may reflect impaired immune surveillance and a higher risk of recurrence [44]. These findings underscore the multifaceted roles of LPC, which vary depending on its concentration, the context of production, the involved cell types, and the specific pathological conditions. Further investigation into these mechanisms may provide new therapeutic strategies for managing inflammation and enhancing antitumor immunity. However, this study has several limitations, including its retrospective design and single-center cohort, which may limit reproducibility. Larger, prospective studies are needed to validate the association between LPC and response to immunotherapy. Additionally, preclinical models could help elucidate LPC metabolism in cancer and inform the development of approaches to augment immune-mediated antitumor effects.

In this study, the differences in prognosis were largely attributable to variations in the efficacy of anti-PD-1 antibody treatments. These findings suggest that ICIs may be ineffective in patients with chronic inflammation, potentially due to a state of immunological exhaustion induced by sustained inflammatory processes. Low LPC levels may reflect a compromised overall immune status, providing a systemic perspective on immune function. An interesting report shows consistency with this concept, as another phenotype of patients who respond to ICIs is those who develop immune-related adverse events. The plasma metabolic profiles of these patients are almost consistent with the metabolites identified in module 6 within our study [45]. Current biomarkers for ICI efficacy are primarily tumor-centered, focusing on factors such as PD-L1 expression, tumor mutational burden, and T-cell infiltration within the tumor microenvironment [46]. However, these biomarkers do not account for the systemic immune capacity of the host to respond to a PD-1 blockade. The LPC level may be a biomarker that indicates systemic immune status, which focuses on responsiveness to ICIs. Indeed, in this analysis, PD-L1 expression and LPC levels were completely uncorrelated, and the two factors were independent. Incorporating biomarkers that assess both tumor and host immune responses could significantly enhance the predictive accuracy of ICI efficacy. Moreover, the implications of LPCs extend beyond prognosis prediction to potential therapeutic applications. Modulating LPC metabolism through targeted interventions may improve cancer therapy outcomes and enhance the efficacy of ICI therapies. Drugs designed to regulate LPC levels could potentially mitigate chronic inflammation and restore immune competence, thereby optimizing responses to anti-PD-1 therapies. Although this study highlights the prognostic and therapeutic relevance of LPCs, the underlying mechanisms connecting LPC decline, poor prognosis, chronic inflammation, and ICI efficacy remain unclear. Future research should aim to elucidate these relationships, providing a mechanistic framework that could inform the development of novel therapeutic strategies.

## 4. Materials and Methods

### 4.1. Clinical Samples and Data

This study included 149 patients with esophageal or head and neck cancer treated at the Department of Medical Oncology, Tohoku University Hospital, between September 2018 and March 2023. All patients had histologically confirmed stage IV SCC. Chemotherapy was administered in accordance with the guidelines of each time period [47,48]. Platinum-based drugs, 5-fluorouracil, nivolumab, pembrolizumab, ipilimumab, and taxanes were mainly used to treat esophageal cancer, while the head and neck cancer treatments were almost identical, but with cetuximab in place of ipilimumab. The median observation period from 1st-line treatment was 601 days. Plasma samples were stored in the hospital clinical biobank. Patient data were retrieved from the electronic medical record system. Patients receiving chemotherapy at the facility were included in this study, and cases with infection at the time of sampling were excluded. Before enrollment, written informed consent was obtained from all participants. Plasma samples were manually aliquoted into storage tubes and stored at −80 °C until further analysis.

### 4.2. Metabolomics Analysis

Wide-target metabolomics analysis was performed using the MxP^®^ Quant 500 XL kit (Biocrates Life Science AG, Innsbruck, Austria) with an ultra-performance liquid chromatography (UPLC) system (ACQUITY UPLC I-Class, Waters Corporation, Milford, MA, USA) coupled to a triple quadrupole mass spectrometer (Xevo TQ-XS, Waters Corporation) as described in prior studies [49]. Briefly, calibration standards, quality control solutions, and plasma samples were prepared and measured according to the kit manual. Metabolite concentrations were calculated from the exported raw files using WebIDQ software (ver. 780, Biocrates Life Science AG). All analyses were conducted at the Tohoku Medical Megabank Organization, and data from healthy individuals (obtained from the jMorp database on the ToMMo website) were also analyzed using the same equipment and methods [50].

### 4.3. Weighted Gene Correlation Network Analysis (WGCNA)

WGCNA was performed to evaluate correlations between metabolites and clinical data [51]. A total of 503 metabolites, detected in >80% of samples, were selected for analysis. Metabolite clustering was conducted using the WGCNA package in R version 1.73. To achieve scale-free topology, the soft-threshold power was determined to be 7 (R^2^ ≥ 0.90). For module detection, the following parameters were applied: TOMType = “unsigned,” minModuleSize = 10, deepSplit = 2, and reassignThreshold = 0. Pearson correlation coefficients were calculated between each module and clinical data, with significance set as a q value < 0.1.

### 4.4. Proteomics Analysis

Proteome analysis was performed using previously reported methods with modifications [52]. Plasma samples were prepared using the ENRICH-iST kit (PreOmics GmbH, Planegg-Martinsried, Germany) following the manufacturer’s instructions. Proteins were extracted and digested, and the resulting solutions were analyzed by liquid chromatography–mass spectrometry using an EASY-nLC 1000 UPLC system (Thermo Fisher Scientific, Waltham, MA, USA) coupled to an Orbitrap Fusion™ Tribrid™ mass spectrometer (Thermo Fisher Scientific). Peptide samples were loaded onto a trap column (Acclaim PepMap 100; Thermo Fisher Scientific) with eluent A, separated on a reverse-phase analytical column (Nano HPLC Capillary column; Nikkyo Technos, Tokyo, Japan), and analyzed using a Nanospray Flex Ion Source system (Thermo Fisher Scientific). Data were acquired in the data-dependent acquisition mode and analyzed using Proteome Discoverer version 3.0 (Thermo Fisher Scientific; RRID:SCR_014477).

### 4.5. Cytokine Assay

Cytokine analysis was performed using the cytometric bead array Human Th1/Th2/Th17 cytokine and human chemokine kit (BD, Biosciences, San Diego, CA, USA) according to the manufacturer’s instructions. This kit allows simultaneous measurement of 12 cytokines and chemokines: interleukins (IL)-2, IL-4, IL-6, IL-10, and IL-17A; tumor necrosis factor (TNF)-α; interferon-γ; and C-C motif chemokine ligands (CCL) 2, CCL5, CXCL8, CXCL9, and CXCL10.

### 4.6. Statistical Analysis

Kaplan–Meier analysis was performed to estimate the distributions of overall survival (OS) in those patients, and a log-rank test was performed to analyze statistical differences in survival. Pathway analysis was performed using the Reactome Pathway Database [53]. Two-tailed *p*-values < 0.05 were considered significant for all tests except proteomic analysis. Proteomic analysis was considered significant if the two-tailed *p*-value was <0.1.

## Figures and Tables

**Figure 1 ijms-26-07528-f001:**
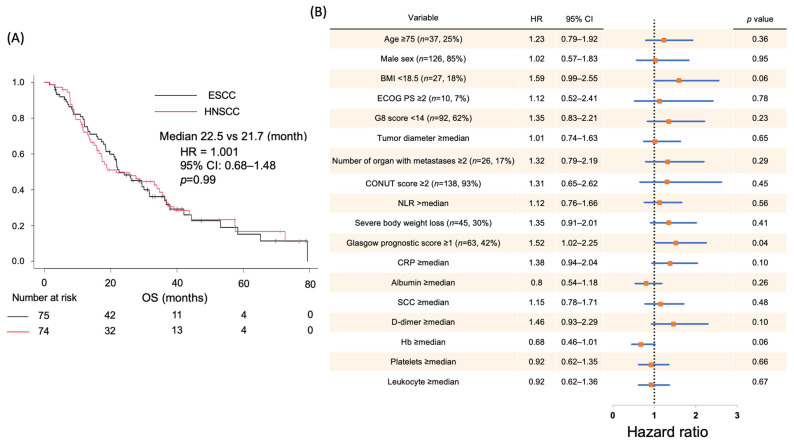
Correlation between clinical data and survival in patients with squamous cell carcinoma (SCC). (A) Kaplan–Meier curve of the overall survival (OS) of patients with either esophageal or head and neck SCC. (**B**) Forest plots of the hazard ratio (HR) for OS associated with clinical data by univariate analyses.

**Figure 2 ijms-26-07528-f002:**
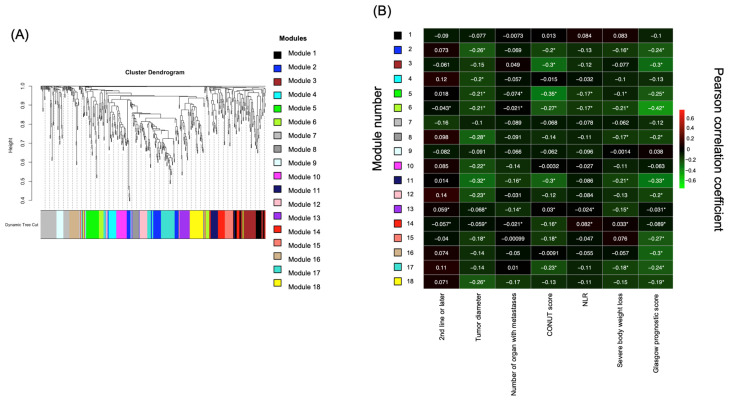
Identification of the metabolite associated with clinical data from the WGCNA. (**A**) WGCNA identified 18 modules, each represented by a unique color. (**B**) Heatmap of the Pearson correlation coefficients between the clinical characteristics and module eigengenes derived from the WGCNA. The asterisks (*) indicate the correlations considered significant (q < 0.1). WGCNA, weighted gene correlation network analysis.

**Figure 3 ijms-26-07528-f003:**
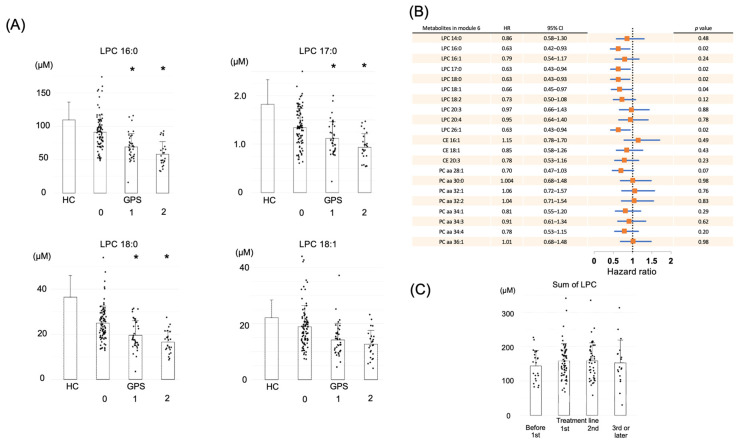
(**A**) Correlation between GPS and major component metabolites in the LPC fraction contained in module 6. * *p* < 0.05 compared with the GPS 0 group. HC, healthy cohort data from jMorp. (**B**) HR for OS associated with each metabolite in module 6. (**C**) Sum of LPC by treatment line at each blood sampling time point. GPS, Glasgow prognostic score; HR, hazard ratio; LPC, lysophosphatidylcholine; OS, overall survival.

**Figure 4 ijms-26-07528-f004:**
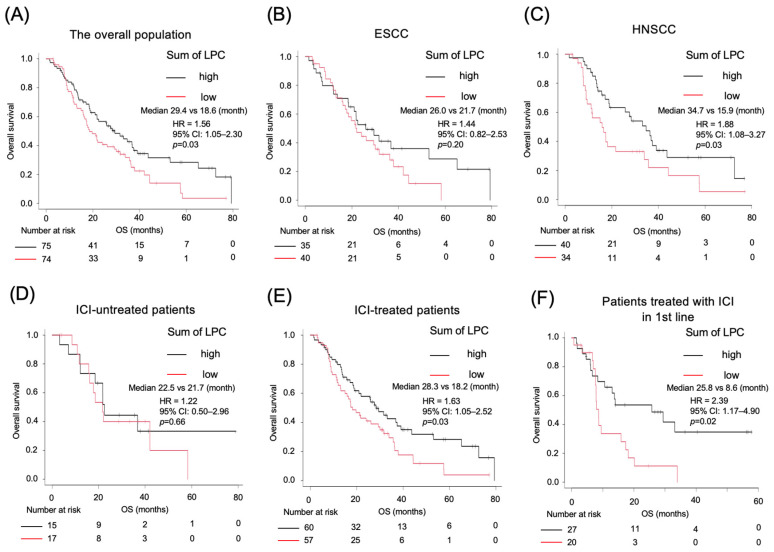
Kaplan–Meier analysis for OS and PFS in different subgroups of patients based on the LPC level. (**A**–**F**) Kaplan–Meier curves of OS between the total high-LPC (≥median) and low-LPC (<median) groups in the overall population (**A**), esophageal SCC group (**B**), head and neck SCC group (**C**), ICI-untreated group (**D**), ICI-treated group (**E**), and ICI-treated group in first-line (**F**). ICIs, immune checkpoint inhibitors; LPC, lysophosphatidylcholine; OS, overall survival; PFS, progression-free survival.

**Figure 5 ijms-26-07528-f005:**
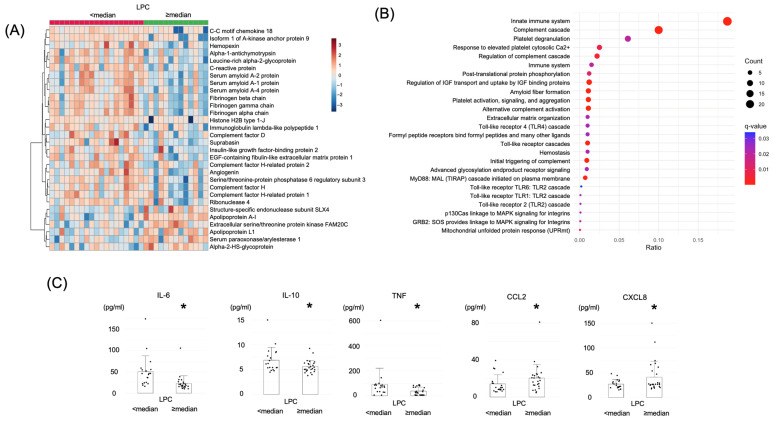
Comprehensive proteome and cytokine/chemokine analyses based on LPC levels. (**A**,**B**) Comparison of the proteomic profiles between the total high-LPC (≥median) and low-LPC (<median) groups. (**A**) Heatmap of the top 30 differential metabolites between high-LPC and low-LPC groups. Metabolites were selected based on fold change and *p*-values. (**B**) Pathway analysis using the Reactome database. (**C**) Comparison of cytokine and chemokine expressions between the total high-LPC (≥median) and low-LPC (<median) groups. * *p* < 0.05. LPC, lysophosphatidylcholine.

**Table 1 ijms-26-07528-t001:** Patient characteristics.

Characteristic	Overall No.	149
Age	Median (range)	69 (31–88)
Sex	Male	126 (85%)
	Female	23 (15%)
BMI, kg/m^2^	Median (range)	21.3 (14.4–34.9)
Body weight loss in past 6 months, *n*	≥5%	42 (28%)
	≥2% and BMI < 18.5	4 (3%)
	Others	103 (69%)
ECOG PS, *n*	0	63 (42%)
	1	74 (50%)
	2, 3	10 (7%)
	Unknown	2 (1%)
Primary lesion, *n*	Head and neck	74 (50%)
	Nasopharynx	7 (5%)
	Oropharynx	8 (5%)
	Hypopharynx	20 (13%)
	Larynx	5 (3%)
	Oral cavity	32 (21%)
	Paranasal and nasal cavity	1 (1%)
	Ear canal	1 (1%)
	Esophagus	75 (50%)
Treatment line at time of blood collection, *n*	Before 1st line administration	21 (14%)
	1st line	64 (43%)
	2nd line	46 (31%)
	3rd line or later	18 (12%)
Tumor diameter, mm	Median (range)	36 (11–143)
Number of organs with metastases	≤1	123 (83%)
	≥2	26 (17%)
CONUT score	Median (range)	4 (0–9)
NLR	Median (range)	3.9 (0.995–66.6)
Glasgow prognostic score	0	86 (58%)
	1	31 (21%)
	2	32 (21%)
CRP, mg/dL	Median (range)	0.46 (0.02–13.95)
Albumin, g/dL	Median (range)	3.8 (2.2–4.8)
SCC, ng/mL	Median (range)	1.80 (0.40–87.3)
D-dimer, μg/mL	Median (range)	1.3 (0.5–23.2)
Hemoglobin, g/dL	Median (range)	12.1 (7.8–17.0)
Platelet, 10^3^/μL	Median (range)	251 (44–800)
Leukocyte, /μL	Median (range)	6400 (1900–15,500)
HDL cholesterol, mg/dL	Median (range)	17.8 (3.4–140.1)
LDL/VLDL cholesterol, mg/dL	Median (range)	91.8 (34.0–329.8)
Total cholesterol, mg/dL	Median (range)	113.0 (49.0–351.7)

Abbreviation: BMI = body mass index. ECOG PS = Eastern Cooperative Oncology Group Performance Status. NLR = neutrophil/lymphocyte ratio.

**Table 2 ijms-26-07528-t002:** Correlations between the GPS and metabolites included in module 6.

	Mean Concentration (SD)			
Metabolites in Module 6	Healthy Cohort	Patients		
		GPS0	GPS1	GPS2
LPC 14:0	1.7 (0.6)	1.4 (0.5)	1.0 * (0.3)	1.0 * (0.5)
LPC 16:0	113.0 (29.4)	91.3 (25.7)	69.3 * (19.9)	58.3 * (18.7)
LPC 16:1	2.9 (1.0)	2.9 (1.3)	2.0 * (0.8)	1.73 * (0.7)
LPC 17:0	1.8 (0.5)	1.3 (0.4)	1.1 * (0.3)	0.94 * (0.3)
LPC 18:0	37.1 (0.2)	25.0 (7.6)	19.6 * (6.5)	16.5 * (4.9)
LPC 18:1	23.0 (7.1)	18.9 (7.5)	14.2 * (5.9)	12.6 * (5.0)
LPC 18:2	33.5 (13.5)	29.4 (12.2)	20.9 * (7.7)	19.0 * (7.7)
LPC 20:3	1.9 (0.7)	1.5 (0.6)	1.1 * (0.5)	1.0 * (0.4)
LPC 20:4	5.9 (2.0)	4.8 (1.9)	4.0 * (1.6)	3.2 * (1.2)
LPC 26:1	0.58 (0.7)	0.32 (0.1)	0.27 * (0.1)	0.25 * (0.1)
CE 16:1	21.1 (17.0)	22.1 (16.0)	15.1 * (9.0)	12.6 * (6.8)
CE 18:1	89.0 (64.9)	108.3 (49.6)	94.2 (56.8)	76.1 * (33.7)
CE 20:3	6.2 (3.9)	5.3 (2.8)	4.8 (2.9)	4.4 (3.0)
PC aa 28:1	3.4 (0.9)	3.4 (1.1)	3.0 * (0.9)	2.9 * (0.9)
PC aa 30:0	3.8 (1.2)	3.8 (1.7)	2.8 * (1.0)	3.1 * (1.5)
PC aa 32:1	13.4 (6.6)	17.8 (15.9)	10.9 * (5.5)	11.4 * (7.3)
PC aa 32:2	3.1 (1.1)	2.8 (1.6)	2.0 * (1.2)	2.0 * (1.6)
PC aa 34:1	199.6 (53.0)	207.2 (82.1)	166.0 * (48.4)	166.8 * (52.7)
PC aa 34:3	13.2 (3.9)	14.1 (6.3)	10.7 * (4.3)	10.3 * (3.7)
PC aa 34:4	1.4 (0.5)	1.2 (0.6)	0.87 * (0.4)	0.79 * (0.4)
PC aa 36:1	45.4 (11.2)	39.4 (13.0)	32.7 * (10.5)	33.2 * (10.2)

Unit: µM. * *p* < 0.05 compared with the GPS 0 group. Abbreviation: GPS, Glasgow prognostic score; LPC, lysophosphatidylcholine; CE, cholesteryl ester; PC, phosphatidylcholine.

## Data Availability

The original contributions presented in this study are included in the article. Further inquiries can be directed at the corresponding authors.

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
