# Peer review of "Plasma Lysophosphatidylcholine Levels Correlate with Prognosis and Immunotherapy Response in Squamous Cell Carcinoma"

_ijms, 2025, doi:10.3390/ijms26157528_

Round 1
Reviewer 1 Report
Comments and Suggestions for Authors
- There are some linguistic problems in the paper that need improvement. Enhancements to the language usage and style are suggested to enhance the overall quality and readability of the manuscript.
- Patients were sampled at different treatment stages (1st-line, 2nd-line, etc.), potentially introducing confounders.
Suggestion: Discuss how treatment timing might affect LPC dynamics or perform subgroup analyses. - Figure 4 shows LPC’s prognostic significance is stronger in head/neck SCC than esophageal SCC. The authors mention adjusting cutoff values but omit details.
Suggestion: Provide supplementary data or discuss potential biological differences (e.g., anatomical site-specific lipid metabolism). - Proteomics analysis uses *p*<0.1 as significance threshold—justify this leniency compared to conventional (*p*<0.05) standards.
- Figure 4 shows LPC’s prognostic significance is stronger in head/neck SCC than esophageal SCC. The authors mention adjusting cutoff values but omit details. Suggestion: Discuss potential biological differences (e.g., anatomical site-specific lipid metabolism).
- In the introduction section, I suggest the authors to discuss the recent updates in the related field.
- Discussion Enhancements:
Address LPC’s dual role in inflammation (pro- vs. anti-inflammatory) and reconcile apparent contradictions.Explicitly state study limitations (e.g., retrospective design, single-center cohort) and advocate for future prospective validation.
There are some linguistic problems in the paper that need improvement. Enhancements to the language usage and style are suggested to enhance the overall quality and readability of the manuscript.
Author Response
Reviewer(s)' Comments to Author:
Reviewer 1
Comments to the Author
- There are some linguistic problems in the paper that need improvement. Enhancements to the language usage and style are suggested to enhance the overall quality and readability of the manuscript.
Although the initial manuscript was thoroughly reviewed by a native English-speaking editor with expertise in academic writing, we have requested an additional round of proofreading through MDPI Author Services in response to the reviewers' comments.
- Patients were sampled at different treatment stages (1st-line, 2nd-line, etc.), potentially introducing confounders. Suggestion: Discuss how treatment timing might affect LPC dynamics or perform subgroup analyses.
As shown in Figure 3C, the samples were collected at different stages of treatment. Importantly, they were obtained during treatment-free periods, minimizing the potential confounding effects of chemotherapy. No significant differences in LPC concentrations were observed between samples from different stages of disease progression. These findings suggest that changes in LPC levels are more likely driven by interactions between cancer cells and the host, rather than by the stage of disease progression. Such interactions may be influenced by tumor characteristics, including rapid proliferation and immunogenicity. This point is addressed in the Results section on page 4. In response to the reviewers’ suggestion, the text has been revised and expanded as follows.
Although samples were collected from patients at various treatment stages (Fig. 3C), no significant differences in LPC levels were observed across treatment lines. This suggests that LPC levels may not serve as a marker of disease progression, but may instead reflect host–tumor interactions rather than treatment stage.
- Figure 4 shows LPC’s prognostic significance is stronger in head/neck SCC than esophageal SCC. The authors mention adjusting cutoff values but omit details.
Suggestion: Provide supplementary data or discuss potential biological differences (e.g., anatomical site-specific lipid metabolism).
To simplify our analysis, we used the median LPC value (LPC = 155) as the cutoff. However, the most pronounced difference in prognosis was observed in the lowest quintile of LPC values (LPC = 105), as shown in Supplementary Figure S2. At this cutoff, significant differences in overall survival were also seen in both the head and neck SCC and esophageal SCC groups. Nevertheless, applying this lower cutoff would result in a smaller number of cases classified as Low LPC, thereby reducing the statistical power of subsequent analyses.
This point has been revised in the main text on page 12 as follows.
In this study, cutoff values were set at the median (LPC = 155) to simplify analysis. How-ever, the greatest difference in prognosis was observed in the lower quintile of LPC values (LPC = 105; Supplementary Figure S2). At this cutoff, significant differences in OS were also observed in both the head and neck and the esophageal SCC groups.
- Proteomics analysis uses *p*<0.1 as significance threshold—justify this leniency compared to conventional (*p*<0.05) standards.
In similar analyses, many studies adopt a significance threshold of p<0.1 to report a broader range of detected proteins. As suggested by the reviewer R1, we reanalyzed the data using a stricter threshold of p<0.05. With this adjustment, the number of identified proteins decreased from 54 to 33. Nevertheless, the analytical pattern remained largely unchanged, and the same innate immune system–related inflammatory proteins were predominantly detected, as shown in the figure for the reviewer R1.
This point has been addressed in the Results section (page 5) as follows.
This result was consistent with the original analytical pattern when a stricter significance threshold of p<0.05 was applied (data not shown).
- Figure 4 shows LPC’s prognostic significance is stronger in head/neck SCC than esophageal SCC. The authors mention adjusting cutoff values but omit details. Suggestion: Discuss potential biological differences (e.g., anatomical site-specific lipid metabolism).
This is the same question as Question 3.
- In the introduction section, I suggest the authors to discuss the recent updates in the related field. Discussion Enhancements:
As pointed out by the reviewers, the introduction has been rewritten to include recent studies and advances in analytical technologies on systemic metabolism, as follows in page 2.
At diagnosis, many patients with SCC present with cachexia, which is a serious deterioration of their general condition that negatively influences the prognosis and complicates treatment. Cancer-related metabolic disruptions extend beyond glycolysis, involving abnormalities in amino acid and lipid metabolism that impair systemic physiological functions. To address these challenges, metabolomic and proteomic analyses have emerged as promising alternatives to genomic approaches. In particular, recent technological advances have enabled the detection of a wider range of metabolites with higher sensitivity. These altered metabolites are implicated not only in the direct regulation of tumor proliferation but also in changes in the tumor microenvironment and activation of immune cells. Elucidating the metabolic cycles and biochemical- and protein-level changes associated with SCC may contribute to the discovery of potential biomarkers, providing insight into the underlying disease mechanisms to aid diagnosis and therapeutic targets. These complementary strategies may help in developing more effective and personalized treatment options.
- Address LPC’s dual role in inflammation (pro- vs. anti-inflammatory) and reconcile apparent contradictions.Explicitly state study limitations (e.g., retrospective design, single-center cohort) and advocate for future prospective validation.
As pointed out by the reviewers, the discussion has been revised to more clearly explain the dual role of LPC in inflammation (page 12-13).
In this study, reduced LPC levels were associated with systemic inflammation in cancer patients. Notably, despite this heightened inflammatory state, ICI therapy was less effective, suggesting a complex interplay between inflammation and antitumor immunity. LPC is classically known for its pro-inflammatory functions, serving as a mediator that activates immune cells and vascular endothelial cells. It is generated through phospholipase A2 activation in response to cellular damage or oxidative stress and is elevated in various inflammatory conditions such as tissue injury, atherosclerosis, infections, and autoimmune diseases. Conversely, LPC also exhibits anti-inflammatory effects under certain conditions. It has been reported to induce immunosuppressive cell populations, such as regulatory T-cells and myeloid-derived suppressor cells, and to promote anti-inflammatory cytokines like IL-10. In chronic inflammatory diseases—including infections, drug-induced pneumonitis, and lipopolysaccharide (LPS)-induced shock—reduced LPC levels have been observed. In cancer, such reductions may reflect impaired immune surveillance and a higher risk of recurrence. These findings underscore the multifaceted roles of LPC, which vary depending on its concentration, the context of production, the involved cell types, and the specific pathological conditions. Further investigation into these mechanisms may provide new therapeutic strategies for managing inflammation and enhancing antitumor immunity. However, this study has several limitations, including its retrospective design and single-center cohort, which may limit reproducibility. Larger, prospective studies are needed to validate the association between LPC and response to immunotherapy. Additionally, preclinical models could help elucidate LPC metabolism in cancer and inform the development of approaches to augment immune-mediated antitumor effects.
In this study, the differences in prognosis were largely attributable to variations in the efficacy of anti-PD-1 antibody treatments. These findings suggest that ICIs may be ineffective in patients with chronic inflammation, potentially due to a state of immunological exhaustion induced by sustained inflammatory processes. Low LPC levels may reflect a compromised overall immune status, providing a systemic perspective on immune function.
Reviewer 2 Report
Comments and Suggestions for Authors
Iwasaki et al have shown that lysophosphatidylcholine could be a biomarker in cancer immunotherapy. Several points should be modified.
- FIist of all, carefully check the manuscript before the submission. In the results section, the comments from editorial office remain in the manuscript Line 71-73. "This section may be divided by subheadings. It should provide a concise and precise description of the experimental results, their interpretation, as well as the experimental conclusions that can be drawn. ".
- Table 1 is difficult to read.
- In line 192, the authors described that a decrease in LPC is linked to immune system activation. Dues sentence this fit with the authors conclusion that low LPC is poor biomarker of ICI?
- If GPS is enough as a biomarker, what is the point of measuring LPC? Is there any advantage of LPC compared to GPS? The authors are advised to perform multivariate analysis including LPC and GPS?
- In line 144, the authors stated that LPCs may not be biomarkers that reflect disease progression. Since LPC is not associated with tumor progression, what makes difference between the patients with high and low LPC?
- Please examine the survival analysis of ICI-treated patients stratifying by the PD-L1 expression (TPS or CPS). Is there any relationship between PD-L1 expression and LPC?
Author Response
Reviewer 2
Comments to the Author
- First of all, carefully check the manuscript before the submission. In the results section, the comments from editorial office remain in the manuscript Line 71-73. "This section may be divided by subheadings. It should provide a concise and precise description of the experimental results, their interpretation, as well as the experimental conclusions that can be drawn. ".
This is our mistake. We have deleted it as the reviewer pointed out. My apologies.
- Table 1 is difficult to read.
As pointed out by the reviewer, we have revised the table to be clearer.
- In line 192, the authors described that a decrease in LPC is linked to immune system activation. Dues sentence this fit with the authors conclusion that low LPC is poor biomarker of ICI?
That is correct. Our data show that ICI treatment is less effective in cases with high systemic inflammatory levels, including CRP. This is thought to be due to immune exhaustion in cancer patients, which is associated with a lack of response to ICI.
This also addresses question 7 from reviewer 1 and is mentioned in the discussion section in page 13 as follows.
In this study, the differences in prognosis were largely attributable to variations in the efficacy of anti-PD-1 antibody treatments. These findings suggest that ICIs may be ineffective in patients with chronic inflammation, potentially due to a state of immunological exhaustion induced by sustained inflammatory processes. Low LPC levels may reflect a compromised overall immune status, providing a systemic perspective on immune function.
- If GPS is enough as a biomarker, what is the point of measuring LPC? Is there any advantage of LPC compared to GPS? The authors are advised to perform multivariate analysis including LPC and GPS?
We greatly appreciate the reviewers' insightful comments. Our primary question is to identify metabolic pathways characteristic of patients with poor prognosis and high GPS. The current data suggest that pathways involving LPC are associated with unfavorable clinical outcomes. We hope that future studies will further elucidate whether alterations in LPC-related metabolic pathways are linked not only to prognosis but also to treatment resistance.
Regarding the multivariate analysis including both LPC and GPS, LPC was identified as the metabolite most strongly correlated with GPS. Consequently, when both variables were included in the multivariate analysis, neither remained statistically significant as independent predictors, likely due to collinearity (data not shown).
- In line 144, the authors stated that LPCs may not be biomarkers that reflect disease progression. Since LPC is not associated with tumor progression, what makes difference between the patients with high and low LPC?
While LPC levels may be influenced to some extent by tumor size or stage, they are generally considered a host-derived factor. LPC is thought to reflect the dynamic interaction between the tumor and the host, potentially influenced not only by tumor burden but also by tumor characteristics such as growth rate and immunogenicity.
As noted in Comment 2 by Reviewer 1, the following revision has been made in page 4.
Although samples were collected from patients at various treatment stages (Fig. 3C), no significant differences in LPC levels were observed across treatment lines. This suggests that LPC levels may not serve as a marker of disease progression, but may instead reflect host–tumor interactions rather than treatment stage.
- Please examine the survival analysis of ICI-treated patients stratifying by the PD-L1 expression (TPS or CPS). Is there any relationship between PD-L1 expression and LPC?
As suggested by the reviewers, we classified PD-L1 expression into negative and positive groups in this analysis and performed survival analysis.
As shown in the figure for reviewer R2, PD-L1 expression in pathological tissue correlated with prognosis (A). However, in this data set, no difference was observed between PD-L1 expression and LPC levels (B). These two factors were independent.
This is addressed in the Discussion section on page 13, which states.
Indeed, in this analysis, PD-L1 expression and LPC levels were completely uncorrelated, and the two factors were independent. Incorporating biomarkers that assess both tumor and host immune responses could significantly enhance the predictive accuracy of ICI efficacy.
Round 2
Reviewer 2 Report
Comments and Suggestions for Authors
The authors have adequately answered my previous concerns.